# Can physiological network mapping reveal pathophysiological insights into emerging diseases? Lessons from COVID-19

Cindy Xinyu Ji[1], Majid Sorouri[2], Mohammad Abdollahi[3], Omalbanin Paknejad[4], Ali R. Mani[1,5]*

1 Network Physiology Lab, Division of Medicine, UCL, London, United Kingdom, 2 Liver and Pancreaticobiliary Research Centre, Digestive Diseases Research Institute, Tehran University of Medical Sciences, Tehran, Iran, 3 Digestive Diseases Research Centre, Digestive Diseases Research Institute, Tehran University of Medical Sciences, Tehran, Iran, 4 Department of Pulmonary Medicine, Shariati Hospital, Tehran University of Medical Sciences, Tehran, Iran, 5 Institute for Liver and Digestive Health (ILDH), Division of Medicine, UCL, London, United Kingdom

* a.r.mani@ucl.ac.uk

## Abstract

Network physiology is a multidisciplinary field that offers a comprehensive view of the complex interactions within the human body, emphasising the critical role of organ system connectivity in health and disease. This approach has the potential to provide pathophysiological insights into complex and emerging diseases. This study aims to evaluate the effectiveness of physiological network mapping in predicting outcomes for COVID-19 patients, using data from the first wave of the pandemic. Routine clinical and laboratory data from 202 patients with COVID-19 were retrospectively analysed. Twenty-one physiological variables representing various organ systems were used to construct organ network connectivity through correlation analysis. Parenclitic network analysis was also employed to measure deviations in individual patients' organ system correlations from the reference physiological interactions observed in survivors. We observed distinct features in the correlation network maps of non-survivors compared to survivors. In non-survivors, there was a significant correlation between the level of consciousness and the liver enzyme cluster, a relationship not present in the survivor group. This relationship remained significant even after adjusting for age and degree of hypoxia. Additionally, a strong correlation along the BUN–potassium axis was identified in non-survivors, suggesting varying degrees of kidney damage and impaired potassium homeostasis in non-survivors. These findings highlight the potential of network physiology as a valuable tool for uncovering complex inter-organ interactions in emerging diseases, with applications that could support clinicians, researchers, and policymakers in future epidemics.

**Data availability statement:** All relevant data are within the manuscript and its Supporting information files.

**Funding:** The author(s) received no specific funding for this work.

**Competing interests:** The authors have declared that no competing interests exist.

## Introduction

The COVID-19 pandemic has profoundly impacted global health, suggesting an urgent need for innovative approaches to understand and manage emerging diseases. COVID-19 was initially recognised for its substantial pulmonary involvement, but it has become clear that the disease affects multiple organs, including the renal, neuronal, and cardiovascular systems, among others [1]. The involvement of multiple systems posed significant challenges in the clinical management of patients during the early stages of the COVID-19 pandemic, particularly in developing the expertise and resources required to manage patients effectively. While epidemiological field studies can rapidly provide information on the risk factors associated with the disease, they do not offer insights into its underlying mechanisms. In contrast, experimental studies can deliver detailed information on the pathophysiology of diseases. However, these investigations typically require time to elucidate the mechanisms that enable healthcare professionals to enhance the clinical management of patients during the initial stages of an epidemic caused by an emerging disease. There is a need for an agile methodology that provides insights into the pathophysiology of complex disorders, guiding the adaptation of resources for the optimal management of patients.

Network physiology is a multidisciplinary field that provides a comprehensive view of the complex interactions within the human body, emphasising the crucial role of organ system connectivity in health and disease [2–5]. Several approaches within this field have been previously described. For instance, the correlation network approach has been used to suggest that disrupted organ network connectivity is linked with poor outcomes in critically ill patients at the population level [6]. Additionally, the parenclitic networks approach can create a network representing organ connectivity at the individual patient level with the potential to predict outcome and response to therapy [7–10]. An important advantage of physiological network mapping is providing physiological insight into a disease process, such as the identification of compensatory mechanisms that are involved during the course of the disease. This is particularly important in complex multiorgan disorders (e.g., sepsis) as well as emerging diseases with unknown pathophysiology (e.g., *Sars-Cov2* infection when it was investigated in 2019).

This study aims to assess the effectiveness of physiological network mapping in determining COVID-19 patient outcomes, based on data from the first wave of the pandemic. By retrospectively analysing this data, we aim to assess whether network physiology might have offered early insights into patient outcomes and improved preparedness for future infectious disease outbreaks.

## Methods

### Patient population

This retrospective study was approved by the Ethics Committee of Tehran University of Medical Sciences (TUMS) and was exempted from the requirement for informed patient consent (Research Ethics Certificate Number: IR.TUMS.VCR.REC.1399.002).

Data were accessed for research purposes on 23 September 2020, and the authors did not have access to information that could identify individual participants during data analysis. Patients with confirmed COVID-19, who were admitted to Shariati Hospital (a tertiary centre affiliated with TUMS) during the first wave of the COVID-19 pandemic, from February 11 to June 20, 2020, were included. These patients were admitted to the ICU or to the acute care ward, as during the first wave of the pandemic, intensive care-level services were extended into the wards through a combination of space conversion and equipment redistribution. The inclusion criterion was a confirmed diagnosis of COVID-19 via RT-PCR. Participants were excluded if were more than 15% clinical or laboratory admission data were unavailable. Upon admission, initial vital signs including respiratory rate (RR), oxygen saturation (O$_2$Sat), heart rate (HR), systolic and diastolic blood pressure (SBP and DBP), and temperature (°C) were recorded. The level of consciousness was assessed by clinicians using an ordinal scale: 1 = coma, 2 = stupor, and 3 = conscious [11]. Additionally, a comprehensive panel of routine laboratory parameters was extracted from patients' medical records, including liver function tests (AST: aspartate aminotransferase, ALT: alanine aminotransferase, ALP: alkaline phosphatase, total and direct bilirubin), hematologic markers (Hb: haemoglobin, WBC: white blood cells, platelet count), blood gas values (pH, PCO$_2$, HCO$_3^-$), inflammatory marker (CRP: C-reactive protein), renal function indicators (Cr: serum creatinine, BUN: blood urea nitrogen), coagulation (INR: international normalized ratio), and electrolyte levels (sodium, potassium). Data on patient survival was extracted from medical records. All data were anonymised before data analysis. Due to the retrospective design, not all patients could be followed for 30 days, and patients were censored at discharge from hospital. During data analysis, no imputation was performed. Analyses were based on available cases, allowing inclusion of participants with partially missing data.

## Correlation network mapping

Correlation Network Analysis was performed to investigate the correlation between biomarkers (representing organ system function) at the population level as described in previous reports [12]. 21 clinical or laboratory variables were extracted from the initially collected data. Correlation networks were generated where nodes represent the physiological variables and edges indicate a significant correlation between two variables. The p-values used to assess the significance of correlations between variable pairs were appropriately adjusted using the Bonferroni correction for all pairwise comparisons. With 21 nodes, a p-value smaller than 0.0002381 was considered statistically significant. The edge thickness of the network illustrates the strength of the Pearson correlation coefficient (r) within the network. Considering the differences in sample sizes and the impact of age on COVID-19 mortality, pair-matching based on age and other variables, such as capillary oxygen saturation, was conducted following the initial general analysis as described [12]. In brief, the pair-matching algorithm involves the following steps:

1. Assign a column X (e.g., age or O$_2$Sat) as the pair-match criteria variable for samples.

2. Calculate the sample size for survivor and non-survivor datasets and assign the smallest dataset as the primary dataset to be pair-matched against. Let us denote this smallest dataset as A and the dataset to be matched against as B.

3. For *i*th sample in A, where *i* represents every sample iteration in the dataset A and B$_{match}$ represents a match to the sample A$_i$, if there is:

   a. 1 exact match, assign A$_i$ to output matrix instance C$_i$ and assign B$_{match}$ to output matrix instance D$_i$, and remove A$_i$ and B$_{match}$ from A and B.

   b. More than 1 exact match, assign A$_i$ to output matrix instance C$_i$ and randomly select 1 of the possible B$_{match}$ choices and assign B$_{match}$ to output matrix instance D$_i$, and remove A$_i$ and B$_{match}$ from A and B.

   c. No exact match, expand the pair-match criteria range by ± 1 or an otherwise specified tolerance value (1 year for age and 1% for O$_2$Sat). For *i*th sample in A, if there is:

i. 1 exact match within the specified range, assign $A_i$ to output matrix instance $C_i$ and assign $B_{match}$ to output matrix instance $D_i$, and remove $A_i$ and $B_{match}$ from A and B.

ii. More than 1 exact match within the specified range, assign $A_i$ to output matrix instance $C_i$ and randomly select 1 of the possible $B_{match}$ choices and assign $B_{match}$ to output matrix instance $D_i$, and remove $A_i$ and $B_{match}$ from A and B.

iii. No exact match, assign a blank value to output matrix instance $C_i$ and output matrix instance $D_i$, and remove $A_i$ from A.

4. Remove all instances of empty values from C and column X, designating C and D as pair-matched datasets.

## Parenclitic deviation

The correlation network provides insights into the patient population rather than individual patients. To explore the potential application of network mapping at the individual patient level based on the correlation maps, parenclitic network mapping was performed to assess how relationships between variable pairs in individual patients deviated from the general trends observed in the reference population [7–10,13]. From the correlation network maps, a variable pair of interest (BUN-potassium) was identified, and the distance (δ) between each individual's data point and the reference regression line derived from the non-survivor group was calculated using the following formula:

$$\delta = \frac{|m \times x - y + c|}{\sqrt{m^2 + 1}}$$

where $m$ and $c$ are the gradient and y-intercept of the orthogonal linear regression line of between BUN and serum potassium level in the reference population, respectively, and x and y are the individual measurements of the variable pairs [10].

## Statistical analysis

All statistical analyses were carried out using STATA (STATA/18.0, Stata Corp, USA) and SPSS Statistics 29 (IBM Corp., USA). Data are reported as mean ± standard deviation (SD). Mann-Whitney U test was performed to compare differences in physiological variable measurements as well as the Parenclitic deviations between survivors and non-survivors. The Chi-square test was used for categorical or ordinal variables. The significance level was set at 0.05 for all statistical tests unless stated otherwise. Univariate Cox regression was used for survival analysis, and the outcome and length of stay (LOS) in hospital were adjusted based on the 30-day mortality cut-off.

## Results

A total of 202 patients were included in this study. During their hospitalisation, 54 patients died (the non-survivors), whereas 148 patients were discharged alive from the hospital. The main characteristics of the studied population are shown in Table 1, and the baseline characteristics of the studied population are summarised in Table 2. In general, non-survivors were older and had reduced capillary oxygen saturation, lower mean blood pressure, and reduced level of consciousness. Creatinine, BUN and BUN/creatinine ratio were higher in non-survivors, along with higher levels of AST (Table 2).

Fig 1 shows the correlation network maps in survivors and non-survivors. The number of edges based on correlation coefficients in the non-survivor group was found to be comparable to that in the survivor group after applying Bonferroni correction (Fig 1). Two main clusters were observed: one cluster reflects liver function (AST, ALT, and ALP), while the others predominantly reflect renal and respiratory functions in adjusting blood pH. In this cluster, serum potassium level was correlated with BUN in non-survivors, while serum potassium was correlated with arterial pH in survivors. Non-survivors also exhibited a significant correlation between the degree of consciousness and the liver enzyme cluster which was not observed in the survivor group. A correlation between WBC count, CRP and platelet count was observed in survivors as

**Table 1. The main characteristics of the studied population.**

| | Survivor (n=148) | Non-survivor (n=54) | p value |
|---|---|---|---|
| **Gender: female/total** (%) | 66/148 (44.6%) | 17/54 (31.5%) | 0.094 |
| **Diabetes: count/total** (%) | 103/148 (69.9%) | 36/54 (66.7%) | 0.691 |
| **Hypertension: count/total** (%) | 85/148 (57.4%) | 33/54 (61.1%) | 0.639 |
| **Age** (years) | 58.7±16.2 | 66.2±15.2 | **0.0011** |
| **Length of hospital stay** (days) | 5.8±5.5 | 11.1±10.2 | **0.0001** |

shown in Fig 1. The number of participants used to construct the correlation network edges in the correlation matrix of the variables analysed in the present study is shown in supporting S1 Table.

Following the matching process for age, the total number of edges observed in the non-survivor group (n=54) was higher than that in the survivor group (n=54), but correlations remained significant between BUN-creatinine, pH-Bicarbonate and ALT-AST in survivors (Fig 2). The pattern of correlations didn't change in non-survivors as shown in Fig 2. We also matched the patients based on their level of hypoxia (as judged by capillary oxygen saturation) and observed a similar pattern (see Supporting S1 Fig).

Since after age matching, the main difference in the network between survivors and non-survivors were in the BUN-potassium, we wondered if the deviation of individual patients' data from this axis is associated with outcome in this patient population. Thus, we calculated parenclitic deviations from these axes in all patients as described [7,8,10]. According to the Mann-Whitney U test, there were increased parenclitic deviations observed in the BUN-potassium axis (0.419±0.334 versus 0.581±0.501, p=0.0458, n=198) when survivors were compared with non-survivors. Univariate Cox regression analysis showed that parenclitic deviation along the BUN-potassium axis was significantly associated with 30-day mortality (Hazard ratio [95% confidence interval] = 1.951 [1.001–3.804], p=0.050, n=198). We did not perform a multivariate Cox regression analysis to assess the independence of these variables from other prognostic factors, due to the low statistical power of the study and the high likelihood of a Type II error.

## Discussion

This project aimed to investigate interactions between physiological biomarkers in the COVID-19 population during the first wave of the recent pandemic, using a correlation network mapping approach and evaluate whether these techniques could provide insights into patient outcomes. We observed distinct features in the correlation network maps of non-survivors compared to survivors. In non-survivors, there was a significant correlation between the degree of conscious-ness and the liver enzyme cluster, a relationship that was not present in the survivor group and remained significant even after adjusting for age and the degree of hypoxia. Additionally, we identified a significant correlation along the BUN–potassium axis in non-survivors.

Two clusters of connections were observed in both survivors and non-survivors' groups, one reflecting renal function and the other reflecting the relationship between liver enzymes and cognitive dysfunction in non-survivors (Figs 1 and 2).

***Renal cluster in the correlation maps:*** Renal complications, such as acute kidney injury, are common in patients with COVID-19 and often manifest as acute tubular necrosis. The underlying mechanisms (whether due to direct viral infection or indirect causes such as pre-renal injury) remain under debate [14]. In the population correlation maps, the renal cluster revealed distinct differences between survivors and non-survivors (Fig 1). In non-survivors, BUN levels were significantly correlated with serum potassium, suggesting varying degrees of kidney damage and impaired potassium homeostasis. This disruption likely reflects renal dysfunction, which can be potentially life-threatening by disrupting potassium homeo-stasis. This finding underscores the importance of close monitoring of renal function and electrolyte balance in high-risk hospitalised patients with COVID-19.

**Table 2. Baseline characteristics of the studied population.**

| Summary table | Survivor | Non-survivor | p value |
|---|---|---|---|
| O$_2$Sat (%) (n) | 91.2±6.0 (147) | 82.3±14.10 (54) | **<0.0001** |
| Mean blood pressure (mmHg) (n) | 98±16 (147) | 90±14 (54) | **0.0004** |
| Respiratory rate (breath/min) (n) | 19±4 (147) | 21±7 (53) | 0.063 |
| Heart rate (beat/min) (n) | 97±20 (147) | 96±19 (54) | 0.7717 |
| Body temperature (°C) (n) | 37.4±1.1 (147) | 37.6±1.1 (54) | 0.4081 |
| Consciousness (arbitrary unit) (n) | Coma=0%, Stupor=2%, Conscious=98% (146) | Coma=4%, Stupor=24%, Conscious=72% (53) | **<0.001** |
| Hb (g/dL) (n) | 12.9±2.7 (144) | 11.4±2.8 (54) | **0.0007** |
| WBC (cells/μL) (n) | 7254±5927 (143) | 8442±6447 (52) | 0.2077 |
| Platelet counts (cells/μL) (n) | 193432±106474 (141) | 150319±76331 (54) | **0.0087** |
| CRP (mg/L) (n) | 53.2±39.1 (135) | 75.0±41.6 (53) | **0.0016** |
| Cr (mg/dL) (n) | 1.33±1.34 (146) | 1.87±1.39 (54) | **0.0001** |
| BUN (mg/dL) (n) | 19.6±16.7 (146) | 39.77±31.02 (54) | **<0.0001** |
| AST (IU/L) (n) | 49.0±51.9 (117) | 122.8±301.1 (45) | **0.002** |
| ALT (IU/L) (n) | 35.1±41.8 (117) | 78.9±230.3 (45) | 0.210 |
| ALP (IU/L) (n) | 200.8±162.3 (113) | 264.2±275.0 (45) | 0.1657 |
| Na$^+$ (mEq/L) (n) | 138.8±3.9 (144) | 137.7±4.1 (54) | 0.1023 |
| K$^+$ (mEq/L) (n) | 4.4±0.5 (145) | 4.8±0.9 (53) | 0.0563 |
| INR (n) | 1.40±0.82 (115) | 1.59±0.90 (51) | **0.0296** |
| Arterial pH (n) | 7.394±0.068 (133) | 7.355±0.116 (54) | **0.0469** |
| Arterial PCO$_2$ (mmHg) (n) | 40.67±9.92 (133) | 39.00±11.03 (54) | 0.3504 |
| Arterial HCO$_3^-$ (mmHg) (n) | 24.85±4.66 (133) | 22.24±7.08 (54) | **0.0069** |

Data are reported as mean±standard deviation unless stated otherwise. Abbreviations: n – sample size; O$_2$Sat – capillary oxygen saturation; Hb – Haemoglobin; WBC – White Blood Cell count; Platelet counts – Platelet count; CRP – C-reactive protein; Cr – Serum creatinine; BUN – Blood urea nitrogen; AST – Aspartate aminotransferase; ALT – Alanine aminotransferase; ALP – Alkaline phosphatase; Na$^+$– Sodium; K$^+$– Potassium; INR – International normalized ratio; Arterial pH – Arterial hydrogen ion concentration; Arterial PCO$_2$ – Arterial partial pressure of carbon dioxide; Arterial HCO$_3^-$ – Arterial bicarbonate concentration.

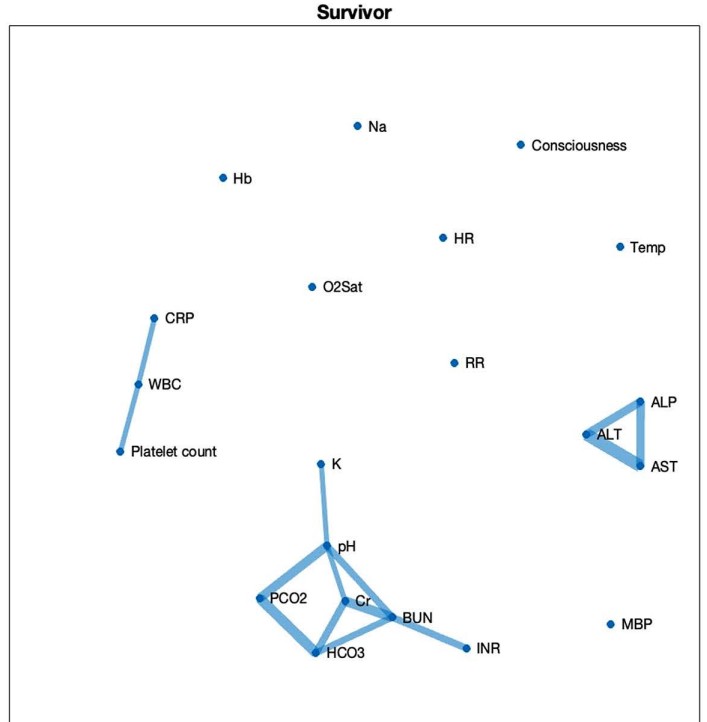
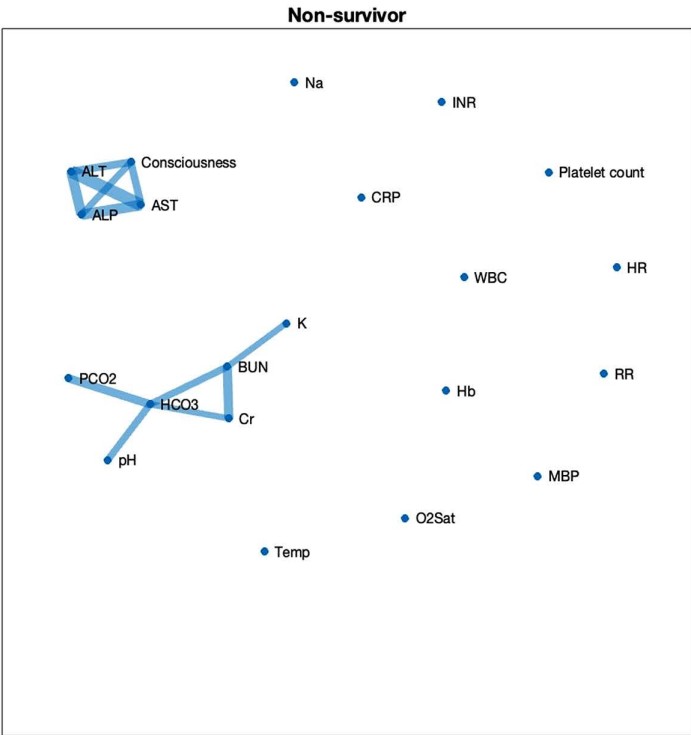

**Fig 1. Bonferroni-Corrected Correlation Network Maps Comparing Survivors (n = 148) and Non-Survivors (n = 54) of COVID-19 during the first wave of the pandemic.** Each link shows a statistically significant Pearson correlation between two biomarkers after Bonferroni correlation for the total number of multiple comparisons. Abbreviations: O2Sat – capillary oxygen saturation; Hb – Haemoglobin; WBC – White Blood Cell count; Platelet counts – Platelet count; CRP – C-reactive protein; Cr – Serum creatinine; BUN – Blood urea nitrogen; AST – Aspartate aminotransferase; ALT – Alanine aminotransferase; ALP – Alkaline phosphatase; Na+ – Sodium; K+ – Potassium; INR – International normalized ratio; pH – Arterial hydrogen ion concentration; PCO2 – Arterial partial pressure of carbon dioxide; HCO3 – Arterial bicarbonate concentration.

In contrast, this BUN–potassium correlation was absent in survivors. Instead, in the survivor group, serum potassium levels were correlated with arterial pH, likely reflecting normal physiological homeostasis, in which excess hydrogen ions move into cells to buffer the pH and are exchanged with extracellular potassium ions to maintain electroneutrality [15]. The difference in the network map regarding plasma potassium correlations suggests a physiological compensatory mechanism in survivors, whereas in non-survivors, the BUN–potassium linkage may reflect a potentially life-threatening disruption. This difference in the renal cluster is important, as it may help avoid nephrotoxic medications in at-risk patients and guide caution when testing emerging antivirals with potential nephrotoxic effects (e.g., remdesivir) at the early stages of the pandemic.

***Liver enzyme cluster in the correlation maps:*** Hepatic involvement in the pathophysiology of COVID-19 is complex and is often associated with mild elevation of liver enzymes, lobular hepatitis and the presence of positive autoantibodies [16,17]. Clinically, while the liver enzyme levels are only mildly elevated in non-survivors, liver function tests have been reported as predictors of mortality in COVID-19 [18]. In the present study, we observed that liver enzymes, especially AST, have consistently shown associations with reduced levels of consciousness in non-survivors. This finding was further validated through Principal Components Analysis, which reduced dimensionality based on the two largest eigenvalues, indicating an association between liver transaminases and consciousness levels in the non-survivor group (see Supporting S2 Table). Our observation on the link between liver enzymes and impaired consciousness in non-survivors could potentially be explained by the liver-brain axis or by urea cycle impairments leading to hyperammonaemia and subsequent

**Fig 2. Age-Matched, Bonferroni-Corrected Correlation Network Maps Comparing COVID-19 Survivors (n = 54) and Non-Survivors (n = 54).** Age matching for the groups was performed using an automated algorithm to pair patients according to their age. Each link shows a statistically significant Pearson correlation between two biomarkers after Bonferroni correlation for the total number of multiple comparisons. Abbreviations: O2Sat – capillary oxygen saturation; Hb – Haemoglobin; WBC – White Blood Cell count; Platelet counts – Platelet count; CRP – C-reactive protein; Cr – Serum creatinine; BUN – Blood urea nitrogen; AST – Aspartate aminotransferase; ALT – Alanine aminotransferase; ALP – Alkaline phosphatase; Na$^+$– Sodium; K$^+$– Potassium; INR – International normalized ratio; pH – Arterial hydrogen ion concentration; PCO2 – Arterial partial pressure of carbon dioxide; HCO3 – Arterial bicarbonate concentration.

encephalopathy [19,20]. However, further studies are needed to confirm the connection between hepatic injury and cognitive function in patients with COVID-19.

Our study used network mapping to explore how network physiology can provide insights into COVID-19 patient outcomes. This approach shows differences in physiological correlations between survivors and non-survivors. Additionally, other recent reports on the application of information theory, principal component analysis, and oxygen saturation entropy analysis in patients with COVID-19 [21–23] further demonstrate the value of network physiology in understanding and managing complex diseases like COVID-19. Such a network-based approach is feasible and can utilise routine clinical, laboratory, and physiological data to generate network maps that offer valuable pathophysiological insights. These insights can enhance our understanding of emerging diseases and support evidence-based policymaking. For example, in the early stages of the pandemic, limited understanding of COVID-19 pathophysiology led to the belief that the disease primarily affected the respiratory system. As a result, policymakers rapidly scaled up the production and deployment of mechanical ventilators and oversaw the impressive and rapid development of large-scale temporal medical facilities (such as the NHS Nightingale Hospitals in the UK) designed to support mechanical ventilation for thousands of patients. However, despite the remarkable speed and scale of construction, the Nightingale Hospitals faced significant limitations and experienced relatively low patient transfer rates [24]. This was largely due to the fact that the care of COVID-19 patients often required complex, multi-disciplinary support, including renal replacement therapies (e.g., haemodialysis, hemofiltration), which were not available in these temporary facilities. Earlier pathophysiological insights (such as recognising

the critical role of renal involvement) could have enabled policymakers to better optimise the design and functionality of emergency healthcare infrastructure.

Likewise, in the early stages of an emerging disease, understanding its pathophysiology is crucial for informed clinical decision-making. Initially, COVID-19 was classified as viral sepsis [25] based on the Sepsis-3 criteria [26]. However, sepsis is a heterogeneous syndrome and involves complex pathophysiology. Our overall connectivity network maps (Fig 2) illustrate no significant reduction in the number of connections observed in the survivor group compared to the non-survivor group. These results contrast with findings from studies on sepsis, which often report that non-survivors exhibit more disrupted organ networks (fewer connections) compared to survivors [6,27,28]. Recent studies have demonstrated distinct disease mechanisms between sepsis and COVID-19, including differences in immune-inflammatory responses and coagulopathy pathways [29]. These differences suggest that COVID-19 and sepsis may involve fundamentally different compensatory mechanisms within the body. Consequently, clinical guidelines developed for the management of sepsis prior to the emergence of COVID-19 may not be fully applicable to critically ill COVID-19 patients (e.g., the use of glucocorticoids or IL-6 monoclonal antibody such as tocilizumab). Therefore, our findings highlight the potential of network physiology to provide unique insights into understanding complex diseases such as COVID-19.

Our study is a retrospective analysis demonstrating the potential of physiological network mapping during the first wave of the COVID-19 pandemic. However, it has inherent limitations, including a patient cohort drawn from a single centre and a specific demographic region. Retrospective clinical studies may be prone to selection bias due to their reliance on existing data and survival analysis derived from medical records. Additionally, medication data as well as other interventions such as mechanical ventilation were not included in the analysis, which could have influenced patient outcomes and the measurement of clinical variables. Since our data reflect the real-life challenges of data collection during the early stages of the pandemic, our dataset is incomplete, with some degree of missing data that could compromise the robustness of the study. While missingness was minimal for more routine data, relatively more transaminase and liver enzyme data (ALT, AST, ALP) values were missing, likely due to logistical difficulties in measuring these parameters for some patients. Therefore, conclusions drawn regarding the contribution of transaminases in this study are less robust and require further investigation using a more well-defined dataset to minimize potential bias due to missingness.

The focus of our investigation is on evaluating the value of correlation network mapping in providing insights into an emerging disease, with the aim of informing policymaking by retrospectively analysing data from the first wave of COVID-19. Correlation maps provide information about the *patient population* as a whole rather than *individual* patients. However, more advanced analytical methods could be applied in the future to further explore physiological network mapping at the individual patient level. One example of such methods is entropy-based analysis, which requires physiological time-series data [28,30] that were not available in this dataset. We explored the potential for parenclitic network mapping in this dataset and found that calculating deviations along the BUN-potassium axis may have promising, though limited, prognostic value. Nonetheless, a full exploration of the prognostic potential of individual network maps requires further work using larger datasets that allow multivariate analysis and validation with independent cohorts. This could be investigated further, alongside novel network-based methods such as Synolytic network mapping [31]. Nonetheless, our findings highlight the promise of network physiology as a valuable tool for uncovering complex interactions between organ systems in emerging diseases. This approach warrants further detailed analysis and the development of network-based methodologies to support clinicians, researchers, and policymakers in future epidemics.

## Supporting information

**S1 Fig. Oxygen saturation-matched, Bonferroni-corrected correlation network maps comparing COVID-19 survivors (n = 44) and non-survivors (n = 44).** Matching for the groups was performed using an automated algorithm to pair patients according to their capillary oxygen saturation (O$_2$Sat). Each link shows a statistically significant correlation between two biomarkers after Bonferroni correlation for the total number of multiple comparisons.

Abbreviations: Hb – Haemoglobin; WBC – White Blood Cell count; Platelet counts – Platelet count; CRP – C-reactive protein; Cr – Serum creatinine; BUN – Blood urea nitrogen; AST – Aspartate aminotransferase; ALT – Alanine amino-transferase; ALP – Alkaline phosphatase; Na$^+$– Sodium; K$^+$– Potassium; INR – International normalized ratio; pH – Arterial hydrogen ion concentration; PCO2 – Arterial partial pressure of carbon dioxide; HCO3 – Arterial bicarbonate concentration.

(TIF)

**S1 Table. The number of participants used to construct the correlation network edges in the correlation matrix of the variables analysed in the present study.**
(XLSX)

**S2 Table. Principal component analysis (PCA) of survivors and non-survivors using Eigenvalue 2 as the cut-off.**
(DOCX)

**S3 Table. Raw data used for network mapping are available as described in the data availability statement.**
Abbreviations: O2Sat – capillary oxygen saturation; MBP – mean blood pressure; RR – respiratory rate; HR – heart rate; T – body temperature; Hb – Haemoglobin; WBC – White Blood Cell count; Platelet counts – Platelet count; CRP – C-reactive protein; Cr – Serum creatinine; BUN – Blood urea nitrogen; AST – Aspartate aminotransferase; ALT – Alanine aminotransferase; ALP – Alkaline phosphatase; Na$^+$– Sodium; K$^+$– Potassium; INR – International normalized ratio; pH – Arterial hydrogen ion concentration; PCO2 – Arterial partial pressure of carbon dioxide; HCO3 – Arterial bicarbonate concentration.
(XLSX)

## Acknowledgments

The authors would like to express their gratitude to all the staff of Shariati Hospital who cared for patients during the COVID-19 pandemic and made the data accessible to the research team for analysis.

## Author contributions

**Conceptualization:** Cindy Xinyu Ji, Majid Sorouri, Ali R. Mani.

**Data curation:** Cindy Xinyu Ji, Majid Sorouri, Mohammad Abdollahi, Omalbanin Paknejad, Ali R. Mani.

**Formal analysis:** Cindy Xinyu Ji, Ali R. Mani.

**Investigation:** Cindy Xinyu Ji.

**Methodology:** Cindy Xinyu Ji, Ali R. Mani.

**Resources:** Majid Sorouri.

**Software:** Cindy Xinyu Ji.

**Supervision:** Ali R. Mani.

**Writing – original draft:** Cindy Xinyu Ji.

**Writing – review & editing:** Cindy Xinyu Ji, Majid Sorouri, Mohammad Abdollahi, Omalbanin Paknejad, Ali R. Mani.

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
