## [Decision Letter · Decision Letter 0]

29 Aug 2025

PONE-D-25-38965Can physiological network mapping reveal pathophysiological insights into emerging diseases? Lessons from COVID-19PLOS ONE?

Dear Dr. Mani,

Thank you for submitting your manuscript to PLOS ONE. After careful consideration, we feel that it has merit but does not fully meet PLOS ONE’s publication criteria as it currently stands. Therefore, we invite you to submit a revised version of the manuscript that addresses the points raised during the review process.

We look forward to receiving your revised manuscript.

Kind regards,

Siddharth Gosavi, MBBS, MD Internal Medicine,DNB Internal Medicine

Academic Editor

PLOS ONE

2. In the online submission form, you indicated that [Data will be made available upon reasonable request.].

Additional Editor Comments:

The study is well written and addresses a clinically relevant question with network analyses of clinical data. Targeted revisions are required to ensure reproducibility, interpretability, and statistical rigour.

1. Network construction and reproducibility

• Correlation networks and the Bonferroni edge-selection rule are described. Specify which correlation coefficient was used (for example Pearson or Spearman), any preprocessing applied before correlation, the missing-data policy, and the effective sample size for each network and each δ-axis. If pairwise deletion was used, report edgewise sample sizes. This allows readers to judge robustness and reproducibility.

• Age-matched and O₂-matched networks are described, and a prior method is cited. For reproducibility in this cohort, add a summary of the implementation: matching approach and ratio, calliper or tolerance used, balance diagnostics (for example pre- and post-match standardised mean differences and the balance criterion), matched sample sizes and any excluded cases, and confirmation that panels labelled as matched used only the matched sets. These choices determine which patients are retained and can change which correlations pass the edge rule.

• Define the admission “consciousness” scale and coding, since it appears in tables and as a δ-axis.

• Make figure captions self-contained by adding sample sizes, correlation type, the edge-selection criterion or threshold, and whether the panel shows unmatched data or which matched set. This improves interpretability for readers consulting figures independently of the text.

2. Survival and prognostic analyses

• Pre-specify multiplicity control and report adjusted p values within families, for example BH-FDR or Holm. Treat the four Mann-Whitney tests as one family and the four Cox tests as a second, and indicate which results remain significant after adjustment. This ensures that reported findings are robust to multiple testing.

• State the time origin for survival analyses, censoring rules, and whether follow-up is complete to 30 days. This is required for reproducibility and correct interpretation.

• δ is analysed in univariable Cox models, with some sensitivity checks (for example age- and O₂-matched networks). These do not replace multivariable adjustment. If an independent prognostic claim is to be retained, run multivariable Cox models for each retained δ-axis, adjusting for a small, pre-specified set of clinical covariates such as age, an oxygenation measure, and a renal function measure. Check the proportional hazards assumption and report hazard ratios with 95% confidence intervals. This establishes whether δ adds prognostic information beyond basic clinical factors.

• If predictive language is to be retained, add internal validation using bootstrap or k-fold methods, and report discrimination and calibration (for example C index or time-dependent AUC at 30 days, plus a calibration plot or Brier score). Clarify whether δ was computed in-sample or against a fixed reference and avoid information leakage by computing δ within folds or using a fixed reference. If validation is not added, temper wording to association and avoid terms such as “predicts” or “independent predictor”.

I congratulate the authors for this research and manuscript which describes the application of a novel network analysis tool (Parenclitic Network Analysis) to the dataset of COVID-19 patients in Iran. Authors have employed this tool to assess the physiological network connectivity/deviation of a group of patients with COVID-19 and the association with survival. Authors reported that overall, COVID-19 patients that survived after 30 days have similar physiological network connectivity compared to those that did not survive although these survivor groups showed different clusters. Specifically, survivors’ serum potassium level which may reflect acid-base balance amongst other physiology is more related to arterial pH in survivors while in non-survivors, the Blood Urea Nitrogen (BUN) is the main driver of serum potassium level with less deviation in survivors along the BUN-potassium axis. I have some comments

Major

1. Methods, Parenclitic deviation section: It is not clear whether the survivors or the non-survivors were used as reference population. Please rephrase or clarify.

Also, in this session, please consider referencing various recent works that have used similar techniques with in-depth description and representation of the parenclitic analysis for clarity.

2. Table 2: It is not clear how consciousness was measured. Please include a definition or description of this in the methodology or as supplementary material.

Also, include the unit of measurement for the variables presented.

3. The authors have not performed any analysis to assess whether the difference in correlation or physiological network deviation is linked with 30-day survivor. Please clarify why as this is a major step that would have confirmed whether the techniques is suitable for predicting mortality.

4. Also, the data on mechanical ventilation is jot

Minor

1. It is not explicitly clear in the methods section about the population that the study population are either patients admitted to the ICU or otherwise. Please make this clear.

2. Please include the full spellings of the abbreviations in the tables below the tables for clarity.

3. Table 1: consider the convention of providing the counts (%) for categorical variables.

Reviewers' comments:

Reviewer's Responses to Questions

**Comments to the Author**

1. Is the manuscript technically sound, and do the data support the conclusions?

Reviewer #1: Partly

Reviewer #2: Yes

2. Has the statistical analysis been performed appropriately and rigorously?

Reviewer #1: No

Reviewer #2: Yes

3. Have the authors made all data underlying the findings in their manuscript fully available?

Reviewer #1: No

Reviewer #2: Yes

4. Is the manuscript presented in an intelligible fashion and written in standard English?

Reviewer #1: Yes

Reviewer #2: Yes

Reviewer #1: The study is well written and addresses a clinically relevant question with network analyses of clinical data. Targeted revisions are required to ensure reproducibility, interpretability, and statistical rigour.

1. Network construction and reproducibility

• Correlation networks and the Bonferroni edge-selection rule are described. Specify which correlation coefficient was used (for example Pearson or Spearman), any preprocessing applied before correlation, the missing-data policy, and the effective sample size for each network and each δ-axis. If pairwise deletion was used, report edgewise sample sizes. This allows readers to judge robustness and reproducibility.

• Age-matched and O₂-matched networks are described, and a prior method is cited. For reproducibility in this cohort, add a summary of the implementation: matching approach and ratio, calliper or tolerance used, balance diagnostics (for example pre- and post-match standardised mean differences and the balance criterion), matched sample sizes and any excluded cases, and confirmation that panels labelled as matched used only the matched sets. These choices determine which patients are retained and can change which correlations pass the edge rule.

• Define the admission “consciousness” scale and coding, since it appears in tables and as a δ-axis.

• Make figure captions self-contained by adding sample sizes, correlation type, the edge-selection criterion or threshold, and whether the panel shows unmatched data or which matched set. This improves interpretability for readers consulting figures independently of the text.

2. Survival and prognostic analyses

• Pre-specify multiplicity control and report adjusted p values within families, for example BH-FDR or Holm. Treat the four Mann-Whitney tests as one family and the four Cox tests as a second, and indicate which results remain significant after adjustment. This ensures that reported findings are robust to multiple testing.

• State the time origin for survival analyses, censoring rules, and whether follow-up is complete to 30 days. This is required for reproducibility and correct interpretation.

• δ is analysed in univariable Cox models, with some sensitivity checks (for example age- and O₂-matched networks). These do not replace multivariable adjustment. If an independent prognostic claim is to be retained, run multivariable Cox models for each retained δ-axis, adjusting for a small, pre-specified set of clinical covariates such as age, an oxygenation measure, and a renal function measure. Check the proportional hazards assumption and report hazard ratios with 95% confidence intervals. This establishes whether δ adds prognostic information beyond basic clinical factors.

• If predictive language is to be retained, add internal validation using bootstrap or k-fold methods, and report discrimination and calibration (for example C index or time-dependent AUC at 30 days, plus a calibration plot or Brier score). Clarify whether δ was computed in-sample or against a fixed reference and avoid information leakage by computing δ within folds or using a fixed reference. If validation is not added, temper wording to association and avoid terms such as “predicts” or “independent predictor”.

3. Data availability

• PLOS ONE requires that data needed to reproduce the findings be publicly available at publication, with rare, justified exceptions. The current “available on request” statement does not comply. Revise the Data Availability Statement to provide a public repository link with a persistent identifier for a de-identified dataset and, where applicable, the analysis code. If legal or ethical constraints apply, describe them, provide a controlled-access route, and make a minimal dataset publicly available. This aligns the manuscript with the journal’s open-data policy and enables verification and reuse.

Reviewer #2: I congratulate the authors for this research and manuscript which describes the application of a novel network analysis tool (Parenclitic Network Analysis) to the dataset of COVID-19 patients in Iran. Authors have employed this tool to assess the physiological network connectivity/deviation of a group of patients with COVID-19 and the association with survival. Authors reported that overall, COVID-19 patients that survived after 30 days have similar physiological network connectivity compared to those that did not survive although these survivor groups showed different clusters. Specifically, survivors’ serum potassium level which may reflect acid-base balance amongst other physiology is more related to arterial pH in survivors while in non-survivors, the Blood Urea Nitrogen (BUN) is the main driver of serum potassium level with less deviation in survivors along the BUN-potassium axis. I have some comments

Major

1. Methods, Parenclitic deviation section: It is not clear whether the survivors or the non-survivors were used as reference population. Please rephrase or clarify.

Also, in this session, please consider referencing various recent works that have used similar techniques with in-depth description and representation of the parenclitic analysis for clarity.

2. Table 2: It is not clear how consciousness was measured. Please include a definition or description of this in the methodology or as supplementary material.

Also, include the unit of measurement for the variables presented.

3. The authors have not performed any analysis to assess whether the difference in correlation or physiological network deviation is linked with 30-day survivor. Please clarify why as this is a major step that would have confirmed whether the techniques is suitable for predicting mortality.

4. Also, the data on mechanical ventilation is jot

Minor

1. It is not explicitly clear in the methods section about the population that the study population are either patients admitted to the ICU or otherwise. Please make this clear.

2. Please include the full spellings of the abbreviations in the tables below the tables for clarity.

3. Table 1: consider the convention of providing the counts (%) for categorical variables.

**Do you want your identity to be public for this peer review?** For information about this choice, including consent withdrawal, please see our Privacy Policy

Reviewer #1: No

Reviewer #2: No

---

## [Author Response · Author response to Decision Letter 1]

13 Oct 2025

Dear Dr Gosavi,

Thank you for handling our manuscript entitled “Can physiological network mapping reveal pathophysiological insights into emerging diseases? Lessons from COVID-19”. We appreciate valuable comments we received from you and the reviewers. We have revised the manuscript accordingly. Below please find our responses to the Editors and reviewer’s comments.

Authors response to the reviewers:

General editorial team comments:

Response: We have revised the manuscript and file names according to PLOS ONE’s style

2. In the online submission form, you indicated that [Data will be made available upon reasonable request.]. All PLOS journals now require all data underlying the findings described in their manuscript to be freely available to other researchers, either 1. In a public repository, 2. Within the manuscript itself, or 3. Uploaded as supplementary information.

Response: Thank you for your suggestion. The data have been uploaded as Supplementary Information (S3 Table) in the revised manuscript.

Reviewer #1:

The study is well written and addresses a clinically relevant question with network analyses of clinical data. Targeted revisions are required to ensure reproducibility, interpretability, and statistical rigour.

1. Network construction and reproducibility

• Correlation networks and the Bonferroni edge-selection rule are described. Specify which correlation coefficient was used (for example Pearson or Spearman), any preprocessing applied before correlation, the missing-data policy, and the effective sample size for each network and each δ-axis. If pairwise deletion was used, report edgewise sample sizes. This allows readers to judge robustness and reproducibility.

Response: Thank you for your comments regarding the writing style and relevance of our manuscript, as well as for your insightful suggestions. We used the Pearson correlation coefficient for constructing the correlation network maps, and this information has now been added to the revised manuscript (Page 5, second paragraph). We have also included additional details on data preprocessing, the handling of missing data, and the effective sample size for each network.

In brief, no data imputation was performed. Analyses were conducted on available cases, allowing the inclusion of participants with partially missing data (Page 5, first paragraph). The number of participants included in each network and in all analyses has been added to Figure 1, Figure 2, Table 1, and Table 2. Furthermore, a new supplementary table (S1 Table) has been added, which shows the number of participants used to construct the correlation network edges in the variable correlation matrices analysed in this study.

Finally, we have added a paragraph in the Discussion section to address potential bias arising from missing data (Page 13, last paragraph; Page 14).

• Age-matched and O₂-matched networks are described, and a prior method is cited. For reproducibility in this cohort, add a summary of the implementation: matching approach and ratio, calliper or tolerance used, balance diagnostics (for example pre- and post-match standardised mean differences and the balance criterion), matched sample sizes and any excluded cases, and confirmation that panels labelled as matched used only the matched sets. These choices determine which patients are retained and can change which correlations pass the edge rule.

Response: Thank you for this comment. We have revised the manuscript to include details of the matching process and the tolerance values used (e.g., ±1 year for age). The number of participants after matching has also been added. Details of the matching procedure can be found in the Methods section on pages 5 and 6 of the revised manuscript.

• Define the admission “consciousness” scale and coding, since it appears in tables and as a δ-axis.

Response: The level of consciousness was assessed by clinicians using an ordinal scale: 1 = coma, 2 = stupor, and 3 = conscious. This has now been added to the method section of the revised manuscript (Page 4).

• Make figure captions self-contained by adding sample sizes, correlation type, the edge-selection criterion or threshold, and whether the panel shows unmatched data or which matched set. This improves interpretability for readers consulting figures independently of the text.

Response: We have updated and revised the figure captions to improve interpretability for readers who consult the figures independently of the main text. Thank you for highlighting this important point.

2. Survival and prognostic analyses

• Pre-specify multiplicity control and report adjusted p values within families, for example BH-FDR or Holm. Treat the four Mann-Whitney tests as one family and the four Cox tests as a second, and indicate which results remain significant after adjustment. This ensures that reported findings are robust to multiple testing.

• State the time origin for survival analyses, censoring rules, and whether follow-up is complete to 30 days. This is required for reproducibility and correct interpretation.

• δ is analysed in univariable Cox models, with some sensitivity checks (for example age- and O₂-matched networks). These do not replace multivariable adjustment. If an independent prognostic claim is to be retained, run multivariable Cox models for each retained δ-axis, adjusting for a small, pre-specified set of clinical covariates such as age, an oxygenation measure, and a renal function measure. Check the proportional hazards assumption and report hazard ratios with 95% confidence intervals. This establishes whether δ adds prognostic information beyond basic clinical factors.

• If predictive language is to be retained, add internal validation using bootstrap or k-fold methods, and report discrimination and calibration (for example C index or time-dependent AUC at 30 days, plus a calibration plot or Brier score). Clarify whether δ was computed in-sample or against a fixed reference and avoid information leakage by computing δ within folds or using a fixed reference. If validation is not added, temper wording to association and avoid terms such as “predicts” or “independent predictor”.

Response: We appreciate your insightful comment and fully agree that survival analysis could substantially strengthen our analysis and the identification of potential prognostic markers. However, we did not perform a survival analysis in the present study, as our primary objective was not to propose a novel prognostic factor, but rather to evaluate the value of correlation network mapping in providing population-level insights into an emerging disease. Our aim was to gain pathophysiologic insight that could potentially inform policymaking by retrospectively analysing data from the first wave of COVID-19.

Correlation maps provide information about the patient population as a whole, rather than individual-level outcomes. Therefore, it would not be appropriate to infer that the characteristics of population-level correlation maps directly identify prognostic markers. Nevertheless, we acknowledge that more advanced analytical methods could be used in future studies to explore physiological network mapping at the individual patient level. Among the available approaches, parenclitic network mapping offers the potential to identify deviations from the population network on an individual basis. In this study, we briefly explored this concept and observed that parenclitic deviation along the BUN–potassium axis was associated with 30-day survival. However, a comprehensive assessment of the prognostic potential of individual network maps will require larger datasets that allow multivariate analyses and validation in independent cohorts. We did not perform multivariate Cox regression in this study due to limited statistical power and the high likelihood of a Type II error. Therefore, the use of parenclitic deviation in this study should be regarded as exploratory, and not as an attempt to identify novel prognostic factors in COVID-19. Our main focus remains on the pathophysiological insights that correlation network maps can provide in the early stages of an emerging disease. In the revised manuscript, we have therefore:

• Retained only the BUN–potassium axis as an illustrative example for future research, removing data from other axes that did not yield robust conclusions. We also emphasised in the method and discussion that the use of parenclitic method is exploratory.

• Clarified that a bootstrap validation (1,000 resamples) of the Cox model for the BUN–potassium deltas indicated stable hazard ratio estimates with minimal bias, although this information was not included in the manuscript given the dataset’s limitations for a full survival analysis.

• Added a note that, due to the retrospective design, not all patients could be followed for 30 days, and patients were censored at discharge from hospital.

• Adjusted the wording throughout to emphasize associations rather than predictions, as you recommended.

• Included these clarifications and limitations in both the Method (page 5), Results (page 10) and Discussion (pages 13–14) sections of the revised manuscript.

3. Data availability

• PLOS ONE requires that data needed to reproduce the findings be publicly available at publication, with rare, justified exceptions. The current “available on request” statement does not comply. Revise the Data Availability Statement to provide a public repository link with a persistent identifier for a de-identified dataset and, where applicable, the analysis code. If legal or ethical constraints apply, describe them, provide a controlled-access route, and make a minimal dataset publicly available. This aligns the manuscript with the journal’s open-data policy and enables verification and reuse.

Response: Thank you for your suggestion. The data have been uploaded as Supplementary Information (S3 Table) in the revised manuscript.

Reviewer #2

I congratulate the authors for this research and manuscript which describes the application of a novel network analysis tool (Parenclitic Network Analysis) to the dataset of COVID-19 patients in Iran. Authors have employed this tool to assess the physiological network connectivity/deviation of a group of patients with COVID-19 and the association with survival. Authors reported that overall, COVID-19 patients that survived after 30 days have similar physiological network connectivity compared to those that did not survive although these survivor groups showed different clusters. Specifically, survivors’ serum potassium level which may reflect acid-base balance amongst other physiology is more related to arterial pH in survivors while in non-survivors, the Blood Urea Nitrogen (BUN) is the main driver of serum potassium level with less deviation in survivors along the BUN-potassium axis. I have some comments

Response: Thank you for carefully reviewing our manuscript and summarizing our results so clearly.

Major

1. Methods, Parenclitic deviation section: It is not clear whether the survivors or the non-survivors were used as reference population. Please rephrase or clarify.

Also, in this session, please consider referencing various recent works that have used similar techniques with in-depth description and representation of the parenclitic analysis for clarity.

Response: We have revised the Methods section and provided further clarification on the use of the parenclitic network, including citations of relevant references. For the calculation of the parenclitic network along the BUN–potassium axis, we used the non-survivor group as the reference, as the correlation was significant in this group. This information can be found in the revised manuscript (Page 6, last paragraph).

2. Table 2: It is not clear how consciousness was measured. Please include a definition or description of this in the methodology or as supplementary material.

Also, include the unit of measurement for the variables presented.

Response: Thank you for this comment. The level of consciousness was assessed by clinicians using an ordinal scale: 1 = coma, 2 = stupor, and 3 = conscious. This has now been added to the method section of the revised manuscript (Page 4). We also added the unit of measurement for all the variables presented in the revised Table 2.

3. The authors have not performed any analysis to assess whether the difference in correlation or physiological network deviation is linked with 30-day survivor. Please clarify why as this is a major step that would have confirmed whether the techniques is suitable for predicting mortality.

Response: You are correct; we did not perform a survival analysis in this study to propose a novel prognostic marker. The focus of our investigation was to evaluate the value of correlation network mapping in providing insights into an emerging disease, with the aim of informing policymaking by retrospectively analysing data from the first wave of COVID-19. Correlation maps offer information about the patient population as a whole rather than about individual patients. Therefore, we cannot directly conclude that the characteristics of population-level correlation maps can lead to the identification of a prognostic marker. However, more advanced analytical methods could be applied in the future to further explore physiological network mapping at the individual patient level. Among the available methods, parenclitic network mapping has the potential to provide information about deviations from the population network at the individual patient level. In this study, we briefly explored this potential and found that the parenclitic deviation along the BUN–potassium axis was associated with 30-day survival. Nonetheless, a comprehensive exploration of the prognostic potential of individual network maps requires further work using larger datasets that allow multivariate analysis and validation with independent cohorts. We did not perform a multivariate Cox regression analysis to assess the independence of these variables from other prognostic factors, due to the low statistical power of our study and the high likelihood of a Type II error. We have suggested that such an analysis be conducted in future studies and have added this limitation in both the Results section (page 10) and the Discussion (pages 13 and 14).

4. Also, the data on mechanical ventilation is jot

Response: You are right. The data on mechanical ventilation at the time of hospital admission were not precise, and therefore we did not include them in this analysis. This limitation has now been acknowledged and discussed in the Discussion section (page 13, second paragraph).

Minor

1. It is not explicitly clear in the methods section about the population that the study population are either patients admitted to the ICU or otherwise. Please make this clear.

Response: Thank you. We have clarified the manuscript and added the following sentence (Page 4, first paragraph): “These patients were admitted to the ICU or to the acute care ward, as during the first wave of the pandemic, intensive care-level services were extended into the wards through a combination of space conversion and equipment redistribution”

2. Please include the full spellings of the abbreviations in the tables below the tables for clarity.

Response: The manuscript was revised according to your comment.

3. Table 1: consider the convention of providing the counts (%) for categorical variables.

Response: Table 1 and 2 were revised according to your comment. Thank you.

---

## [Editor Report · Decision Letter 1]

6 Nov 2025

Can physiological network mapping reveal pathophysiological insights into emerging diseases? Lessons from COVID-19

PONE-D-25-38965R1

Dear Dr. Mani,

We’re pleased to inform you that your manuscript has been judged scientifically suitable for publication and will be formally accepted for publication once it meets all outstanding technical requirements.

Kind regards,

Siddharth Gosavi, MBBS, MD Internal Medicine,DNB Internal Medicine

Academic Editor

PLOS ONE

Additional Editor Comments (optional):

Thank you for your response.I am happy personally with the justifications you have provided.
---

## [Editor Report · Acceptance letter]

PONE-D-25-38965R1

PLOS ONE

Dear Dr. Mani,

I'm pleased to inform you that your manuscript has been deemed suitable for publication in PLOS ONE. Congratulations! Your manuscript is now being handed over to our production team.

Kind regards,

on behalf of

Dr. Siddharth Gosavi

Academic Editor

PLOS ONE